# A Novel Hybrid Deep Learning Model for Human Activity Recognition Based on Transitional Activities

**DOI:** 10.3390/s21248227

**Published:** 2021-12-09

**Authors:** Saad Irfan, Nadeem Anjum, Nayyer Masood, Ahmad S. Khattak, Naeem Ramzan

**Affiliations:** 1Department of Computer Science, Capital University of Science and Technology, Islamabad 44000, Pakistan; mcs191026@cust.pk (S.I.); nayyer@cust.edu.pk (N.M.); 2Department of Computer Science, COMSATS University, Islamabad 45550, Pakistan; ahmad.saeed@cuivehari.edu.pk; 3School of Computing, Engineering and Physical Sciences, University of the West of Scotland, Paisley PA1 2BE, UK; Naeem.Ramzan@uws.ac.uk

**Keywords:** human activity recognition, transition activities, hybrid models, deep learning

## Abstract

In recent years, a plethora of algorithms have been devised for efficient human activity recognition. Most of these algorithms consider basic human activities and neglect postural transitions because of their subsidiary occurrence and short duration. However, postural transitions assume a significant part in the enforcement of an activity recognition framework and cannot be neglected. This work proposes a hybrid multi-model activity recognition approach that employs basic and transition activities by utilizing multiple deep learning models simultaneously. For final classification, a dynamic decision fusion module is introduced. The experiments are performed on the publicly available datasets. The proposed approach achieved a classification accuracy of 96.11% and 98.38% for the transition and basic activities, respectively. The outcomes show that the proposed method is superior to the state-of-the-art methods in terms of accuracy and precision.

## 1. Introduction

Human Activity Recognition (HAR) deals with the recognition, interpretation, and assessment of human daily-life activities. Wearable sensors such as an accelerometer, gyroscope, depth sensors etc. can be attached on assorted body locations to record movement patterns and actions. In recent years, HAR research has attracted critical consideration on account of its boundless applications, such as in fashion [1], surveillance systems [2,3], smart homes [4] and healthcare [5].

Several HAR systems have been designed to automate the aforementioned applications; however, assembling a completely automated HAR framework can be an extremely daunting undertaking since it requires a colossal pool of movement data and methodical classification algorithms. In addition, it is a troublesome undertaking in light of the fact that a solitary movement can be performed in more than one way [6].

Human activities are generally categorized as basic activities, complex activities and the postural transitions between or within these activities. Postural transition is a finite movement between two activities, which varies between humans in terms of time and actions. Most of the works do not take into account the postural transitions because of their short duration. However, when performing multiple tasks in a short period of time, they play an important role in effectively identifying activities [7].

Customary AI tools such as machine learning (M.L) algorithms have been utilized for classification and are capable of achieving satisfactory results. Activity recognition using standard M.L approaches such as K Nearest Neighborhood (KNN) [8,9,10], Support Vector Machine (SVM) [11,12,13,14], Decision Tree (DT) [15,16], Random Forest (RF) [17,18] and Discrete Cosine Transform (DCT) [19,20,21], etc., have been reported to produce good results under controlled environments [22]. The accuracy of these models heavily depends on the process of feature selection/extraction.

A volume of research has been conducted on activity recognition based on features obtained from a variety of datasets collected using various sources such as accelerometers and gyroscopes. However, it is of great importance that a feature selection preprocess module is applied to select a subset that prunes away any noise and redundancy, which would otherwise only degrade the performance of the recognition system. This computation is also called dimensionality reduction; i.e., the selection of features that would complement each other. An assortment of feature selection strategies have been utilized to improve the performance of activity recognition systems, such as correlation-based [23] and energy-based methods [24], cluster analysis [25], AdaBoost [26], Relief-F [27], Single Feature Classification (SFC), Sequential Forward Selection (SFS) [28] and Sequential Floating Forward Search (SFFS) [29]. The aim of feature selection methods is to drop features that carry the least information for discriminating an activity, consequently increasing efficiency without compromising robustness. For more on feature selection methods, readers are referred to a survey in [30]. Mi Zhang and Alexander A.Sawchuk have proposed an SVM-based framework to analyze the effects of feature selection methods on the performance of activity recognition systems [31]. This research has shown that the SFS method performs better than Relief-F and SFC methods. Essentially, this points to the fact that the most relevant features encode more information than the left out features, which considerably increases the performance of the activity recognition system.

Likewise, Ahmed et al. [32] demonstrated a feature selection model based on a hybrid SFFS feature selection method that selects/extracts the best features in view of a set of specific rules. Moreover, sets of the best features are formed and contrasted with the next set of features. The final optimal features were input to a SVM classifier for activity classification. However, machine learning techniques to date have used shallow-structured learning architectures that only use one or two nonlinear layers of feature transformation. In addition, shallow architectures usually refer to statistical features such as the mean, frequency, variance, amplitude, etc., that could only be used for low-level activities such as running, walking and standing that are well-constrained activities and cannot model complex postural situations [33]. Moreover, the lack of good quality data due to the costly process of labeling that requires human expertise/domain knowledge is also a bottleneck, as well as the manual selection of features, which is vulnerable to a margin of human error that would not generalize well to unseen data, because activity recognition tasks in real-life applications are much complicated and require close collaboration with the feature selection module [34].

As the volume of datasets has increased to an unprecedented level, in recent years, deep learning (D.L) has accomplished noteworthy results in the space of HAR. One of the impactful aspects of deep learning is the automatic feature identification and classification with high accuracy, which consequently produced an effect in the space of HAR [35]. A substantial amount of uni-model and hybrid approaches have been introduced to gain benefit from deep learning techniques, catering for the shortcomings of the machine learning domain and utilizing the multiple levels of features found in different levels of hierarchies. Deep learning models involve a hierarchy of layers to accommodate low and high-level features as well as linear and nonlinear feature transformations at these levels, which helps in learning and refining features. To this end, models such as Recurrent Neural Networks (RNN) [36], Convolutional Neural Networks (CNN) [37], Long Short-Term Memory (LSTM) [38], etc., are used to overcome the impediments of traditional M.L algorithms that were dependent on manual features, in which the erroneous selection/classification of features could have undesirable impacts on the applications at hand. Therefore, deep learning networks have found a natural application in recognition tasks and have been popularly used for feature extraction in activity recognition research [39]. One downside to the deep learning paradigm, especially using the hybrid architectures, is their increased cost of processing the available huge amount of datasets. However, it is worth the cost because a HAR system requires accurate classification results of the deep learning models.

In this context, this work proposes a hybrid deep-learning based approach where models are trained simultaneously, instead of in a pipelined setup, to recognize basic and transitional human activities. The novel aspects include the simultaneous implementation of multiple deep learning models to better distinguish the classification results and the inclusion of transitional activities to present a robust activity recognition approach.

The rest of this paper is structured as follows: Section 2 provides the details about existing related works, Section 3 presents the proposed approach, Section 4 discusses the experimentation and results, and Section 5 draws conclusions. Appendix A contains the repository link for the source code and datasets used in this approach.

## 2. Related Works

We have classified the current literature into two groups: approaches based on (Section 2.1) basic activities and (Section 2.2) transition activities. The details are provided in the subsequent subsections.

### 2.1. Basic Activities

Wan et al. [40] demonstrated a CNN framework that showed that the fine-tuned conventional CNN still outperforms SVM, Multilayer Perceptrons (MLP), LSTM and Bidirectional LSTM (BiLSTM) networks. Moreover, the results showed a significant increase in classification accuracy compared to machine learning classification models such as, DT, RF, etc. However, these approaches have limitations as they can only extract simple features. Zhou et al. [41] demonstrated a deep learning framework for weakly labeled data which was able to extract features well. The framework mainly accommodated an auto-labeling technique that was carried out on top of a HAR framework and employed a distance-based reward rule strategy to label the data. The newly labeled data were fused with the strongly labeled data and passed through a LSTM module for feature extraction. This approach was focused on the labeling of unlabeled and weakly labeled data rather than the classification accuracy. To that end, it required a large dataset of unlabeled data for accurate labeling, which resulted in an increased computational cost.

Chen et al. [42] demonstrated an Attention-Based BiLSTM (ABiLSTM) framework by introducing the concept of attention, which assigns weights to features based on their importance for the current recognition scenario. The results showed superior classification accuracy compared to modern approaches in the class of shallow as well as deep learning architectures. All the experimental evaluations were based on the publicly available pre-processed data, and no real-time data collection was performed, which assumes a significant part in the assessment of a signal-based system. Zhu et al. [43] demonstrated a Deep LSTM (DLSTM) architecture for feature recognition and filtration. Smart phone sensors were employed to train the model on labeled and unlabeled data for human activity recognition. DLSTM encapsulated multiple LSTM layers between the I/O gates. The raw data were processed through the augmentation module to build the measure of information, and the removal of Gaussian noise was initiated to filter irregularities in the final input. The DLSTM separated the low-level features, which were dropped out, and the high-level features, which were extracted. The unsupervised data loss affected the unlabeled data was calculated and labeled in light of a number of rules. The benchmarks of the proposed DLSTM, on the publicly available dataset, showed superior results compared to cutting-edge semi-supervised learning frameworks in a user-controlled environment.

Xu et al. [44] fused a conventional RNN with an Inception Neural Network (INN) model targeted at HAR based on wearable sensors to create InnoHAR. The INN architecture is composed of various deep layers consisting of multiple convolution layers that are parallel to pooling layers, forming the inception layer. The INN architecture has been tested on multiple publicly available datasets and it portrayed superior performance compared to Deep–Convolutional–LSTM models. The drawback of this framework was the poor initialization of INN, which required a great deal of computation to be dealt with, and minor changes could require the costly retraining phase to be repeated.

### 2.2. Transition Activities

All of the aforementioned works have a significance of their own; however, all of them are based on basic human actions. None of these works have discussed or employed transition activities. Transition activities (or postural transitions) were publicly introduced in [45]. Although postural transitions may not have an emphasizing effect on the system due to their short duration and lower incidence, the validity of this statement is dependent on the application prospects. Shi et al. [46] proposed a standard deviation trend-analysis (STD-TA)-based architecture to recognize transition activities. For the dimensional reduction of features, only statistical features were extracted, and a conventional SVM was utilized for classifier training. The self-collected dataset was based on 8 basic daily life activities and 10 transitions.

Liu et al. [47] introduced a housekeeping task monitoring system to lay out the significance of activity transition events in housekeeping tasks related to elderly people. The self-generated dataset was divided into three parts which contained the basic housekeeping activities, inter-transition and intra-transition activities. The approach was based on a SVM model with an embedded transition event detection module, and it managed to achieve a classification accuracy of 81.62%. Ahmad et al. [48] proposed a Deep Belief Network (DBN)-based approach to extract features such as mean, median, auto-regressive coefficients, etc., from the raw data obtained from sensors. To make the features more robust, these features were further filtered through a Kernal Principal Component Analysis (KPCA) and Linear Discriminant Analysis (LDA) unit. The proposed model was compared with SVM and ANN networks and was shown to achieve an accuracy of 95.8%.

Gusain et al. [49] proposed a transition-aware Gradient Boosted Decision Tree approach. They implemented incremental learning by utilizing ensembles of SVM. Batches of data were trained on frequent iterations, but after the initial cycle of training, all the other cycles were trained on the incorrectly classified data. Finally, the weighted sum of all the machines was calculated, and the accuracy of the whole system was computed—the accuracy of the whole system was calculated to be 94.9%. Yulita et al. [50] presented a hybrid model based on a classic KNN and SVM model where the SVM kernels were polynomial. Moreover, they combined their approach with the Radial Basis Function (RBF) and Sigmoid function. After cross validation, they managed to achieve an accuracy of 86%. These results were achieved due to the RBF kernel, as it is a useful function to solve classification problems by finding non-linear classifiers. Atrsaei et al. [51] designed a location-independent postural transition detection algorithm. Postural transitions were detected by the sensor following the vertical acceleration calculation, and kinematic features were extracted to characterize the postural transitions. The approach was focused on the algorithm rather than the accuracy of the system. The proposed approach was independent of the placement of sensors on the body and produced satisfactory results. Dan Setterquist [52] evaluated multiple networked LSTM units on a collected dataset of basic activities and postural transitions and managed to achieve 89% accuracy in a user-controlled environment. However, the utilization of multiple LSTM units in a pipelined flow abruptly slows down the whole model and increases the complexity of the system.

In a more recent study, Wang et al. [53] proposed a hybrid D.L. approach in which multi-sensor data were passed through a CNN and the output was classified by the LSTM. The fundamental accomplishment of this approach is the activity transition identification alongside basic activities while emphasizing the fact that most of the research works do not employ the postural transitions; however, in human behavior recognition, this is non-negligible and thus an important task to consider. The accuracy of real-time movement recognition is strongly dependent on the detection of postural transitions. The triaxial multi sensor data were fused and fed to the multilayered CNN. The resultant feature matrix from the CNN was flattened and input to the LSTM module. LSTM was separately trained on the sensor data, and a feature fusion was performed before the final classification. The benchmark showed superior results compared to state-of-the-art CNN, LSTM, CNN-BiLSTM and CNN-GRU models on a publicly available dataset.

Taking into consideration the state-of-the-art technologies and innovations, the M.L. trend is shifting from the traditional high-power consuming hardware devices to low-power mobile devices. This shift is referred to as TinyML [54]. This largely breaks the high-power consumption barrier in the areas of M.L. By focusing on low-power devices, the responsiveness of the whole system can be increased while reducing the power consumption-based cost of the system. Banbury et al. [55] employed a differential neural architecture search (DNAS) to bring forward a MicroNet model deployed on MCU, which showed superior results on TinyML benchmark tasks which included audio-based keyword spotting, visual wake words and anomaly detection. DNAS models were utilized due to their characteristics of requiring low MCU memory and energy.

The current limitations of TinyML are restricted to shrinking the size of the M.L. model; however, with the passage of time and advancements in technology, lightweight neural networks are being designed that can take up to few hundred KBs of space on the TinyML devices and produce substantial results. In future, TinyML could also play an important part in the applications of Augmented Reality (AR) headsets that need to be kept powered on due to shared constraints. For such applications, sensors, such as accelerometer or compass, are used in conjunction with other sensors e.g. gyroscope, heart-rate sensors etc. Though the huge pool of data needs to be segmented for such applications but by having data from multiple sensors, a more interactive and efficient AR environment can be visualized [56].

The discussed literature is summarized in Table 1. The literature signifies the implementation of deep learning networks over conventional machine learning algorithms where accuracy and vast pools of data are a major concern. To develop an efficient and scalable HAR system, this paper introduces a novel hybrid model which takes both basic activities and postural transitions (transition activities) into account. Accordingly, we integrated multiple deep learning models for feature extraction and proposed a decision fusion module for activity recognition.

## 3. Proposed Approach

The architecture for the proposed approach consists of three deep learning (D.L) networks: LSTM, BiLSTM and CNN as depicted in Figure 1. Three D.L. networks are utilized due to the imposition of the decision fusion module in the proposed approach. The proposed approach requires at least three D.L. networks to distinguish between the individual model results and implement decision fusion in an efficient manner. The raw sensor data are converted into a feature matrix and fed to these models separately. Batch normalization is employed in all three networks to normalize the output of each layer [57]. After the classification results are retrieved from each model, a decision fusion module is initiated for the final classification. The subtleties of the proposed approach are given in the accompanying sections.

### 3.1. Long Short-Term Memory (LSTM)

The LSTM framework integrated in this approach is a standard unit contrived of an input gate, output gate, a forget gate and a memory cell. The LSTM unit is graphically shown in Figure 2. The feature matrix is transformed into a 1D vector of *y* elements and fed to the model for training, and the number of neurons are configured to be η. “Adam” is configured to be the adaptive optimizer as it performs best with sparse data. Moreover, the learning rate of μ is adapted to achieve the best results while avoiding the loss of training input. A dropout rate of κ is used to avoid over-fitting while maintaining the integrity of the input and output of neurons. Batch normalization is used after the fully connected layer to normalize the input of every layer in the model, and the Softmax layer classifies the results. Table 2 shows all the hyper parameters in LSTM and their respective values for the two datasets involved.

$f_{t}$ represents the forget gate, which handles the amount of information to be kept and dropped. It is consumed by the sigmoid function, which scales the values between 0 and 1, thus dropping values <0.5. $C_{t}$ represents the input gate that quantifies the importance of the next input ($X_{t}$) and updates the cell state. The new input ($X_{t}$) is standardized between −1 and 1 by the tanh function, and the output is point-wise multiplied by $C_{t}$. $C_{t-1}$ represents the state of the cell at previous timestamps, which is updated after each time step. The information required to update the cell’s state is gathered at this point, and a bit-wise multiplication is carried out between the previous cell state ($C_{T}$) and the forget vector. This is followed by the bit-wise addition with the output of input gate, and the cell state is updated. Finally, the output gate ($O_{t}$) determines what the next hidden state should be. The hidden state encapsulates the information regarding previous inputs ($H_{t-1}$). The flow of information through the following gates is mathematically shown in Equation (Equation 1).
(1)$$\begin{array}{c} f_{t}=sigmoid\left(X_{t}*U_{f}+H_{t-1}*W_{f}\right)\\ C_{t}=tanh\left(X_{t}*U_{C}+H_{t-1}*W_{C}\right)\\ O_{t}=sigmoid\left(X_{t}*U_{O}+H_{t-1}*W_{O}\right) \end{array}$$
where *W* and *U* represent the weights corresponding to their respective gates.

### 3.2. Bidirectional Long Short-Term Memory (BiLSTM)

The BiLSTM model utilized is based on dual recurrent (LSTM) layers, as shown in Figure 3. The top-most layer is referred to as the embedding layer, which predicts the output for different time steps. The second layer is the forward LSTM (first recurrent) layer, which takes the input in the forward direction. The third layer is the backward LSTM (second recurrent) layer, which moves the input in the backwards direction. The first recurrent layer runs the input from the past to future, while the second recurrent layer runs the input from the future to past. The second recurrent layer is provided with the reverse sequence of the input that preserves the future information. This effectively increases the information required by the network for accurate predictions, thus improving the context available to the BiLSTM network. The additional training of data in BiLSTM model shows better results compared to those of LSTM. The BiLSTM hyper parameters were kept the same as those of the LSTM to avoid any inconsistencies in the network and to track the changes in performance on multiple datasets.

### 3.3. Convolutional Neural Networks (CNN)

In this research work, a 2D-CNN is designed which takes its input as a feature matrix ‘‘I". The CNN is comprised of two stacked hidden layers, a fully connected layer, batch normalization layer and a softmax layer for classification, as shown in Figure 4. Each hidden layer is a stack of “Convolution–ReLu–Maxpool” layers. The convolution layer outputs a feature map which is passed through the “Rectified Linear Unit (ReLu)” piece-wise linear function (ϱ). The output of ReLu becomes the input to the pooling layer. Among various sorts of pooling techniques, max pooling chooses the greatest component from each block in the feature map. The pool size deals with the block to be covered and is kept as γ. Padding is set to “SAME”, and the stride is set as ‘‘s", such that the whole input block is covered by the filter.

The numbers of kernels in the two convolution layers are α and β, respectively. The kernel sizes are τ and ν, respectively. Zero padding is added to fill the edges of the input matrix, and a learning rate of ζ is adopted. The input to the convolution layer is of size hxwxd, where *h* represents the height of the input, *w* represents the width of the input, and *d* refers to the dimension of the input. In this approach, the dimension of the input is 0 as we are dealing with sparse sensor data. The convolution layer applies a filter of size $f_{h}xf_{w}xd$, where $f_{h}$ denotes the filter height and $f_{w}$ represents the filter width. The convolution layer outputs a volume dimension or feature matrix ($f_{m}$) as shown in Equation (Equation 2).
(2)$$f_{m}=\left(h-f_{h}+1\right)*\left(w-f_{w}+1\right)*1$$

A batch normalization layer is utilized after the fully connected layer to normalize the data in all the previous layersm, and the output is sent to the softmax layer for classification. The mean and variance calculation in batch normalization is shown in Equations (Equation 3) and (Equation 4), where *x* denotes the batch sample, $\mu_{B}$ represents the batch mean, and $\sigma_{B}^{2}$ represents the mini batch variance. Table 3 shows all the parameters involved in CNN and their respective values.
(3)$$\mu_{B}=\frac{1}{m}\sum_{i=1}^{m}x_{i}$$
(4)$$\sigma_{B}^{2}=\frac{1}{m}\sum_{i=1}^{m}{\left(x_{i}-\mu_{B}\right)}^{2}$$

### 3.4. Model Implementation

An input is fed to each model separately, and activities are classified based on their respective labels. LSTM is utilized for its ability to achieve superior results in sequence to sequence classification. The LSTM encapsulates an input layer, hidden layers and a feed-forward output layer. The hidden layer confines memory cells and multiple gated units. The gated units are divided into input, output and forget gates. A feature vector is fed to the input gate which covers the update gate. The update gate is a combination and works on the same principals as the input and forget gate; thus, it decides which values to let through and which to drop. The *tanh* layer makes sure the values are scaled between −1 and 1. The forget gate sorts out how much information can be aggregated from the previous gate into the memory cell. The *Sigmoid* activation is used to scale the output from the gates between 0 and 1 to speed up the training and reduce the load on network. The results of the output gate are generated based on the cell’s state and flattened by the fully connected layer. All the parameters and inputs to layers are scaled and standardized by the batch normalization layer, and the final output is classified by softmax.

The feature vector is passed to a BiLSTM network. BiLSTM has same parameters and works on the same principles as an LSTM. The only point of difference is that in BiLSTM, the input is fed to the model twice for training: once from beginning to the end and once from end to the beginning. Therefore, by utilizing BiLSTM, we can preserve information at any time at a point in future and past, which generates a refined feature map. Furthermore, BiLSTM speeds up the training process, and this dual training of data better classifies the activities compared to LSTM. The final feature map is flattened and classified by softmax.

For the precise conversion of data into a matrix for CNN, zero padding is added to the input. Convolutions are performed on the matrix and weights are distributed among the filters. A bias is set to update the values of weights after a complete iteration. The output from the convolution layer is passed through the ReLu to convert all the negative values in the resulting feature matrix to zero.

The output of ReLu is input to the pooling layer to shrink the feature matrix and normalize the overall parameters in the hidden layer. Among several pooling techniques, maxpool is the most effective while dealing with sensor data and is thus utilized in our approach.

The feature map from the maxpool layer is input to the second hidden layer, and the whole process is repeated twice. The final feature matrix is flattened in the fully connected (FC) layer and forms a pre-classification sequence. After the FC layer, the batch normalization layer is used to normalize the output of all the layers in CNN. The softmax layer predicts polynomial probability distributions and generates categories based on these predictions.

Softmax is utilized in all models due to its ability to generate statistical probabilities alongside classes. These probabilities are exerted in the decision fusion module for final classification. After all three networks, the observations (instances) corresponding to each activity are input to the models, and each network returns the predicted classes along with the class probabilities. The predictions are then inter-compared and summed in a decision fusion module, and final classification results are generated.

### 3.5. Decision Fusion

The decision fusion module prioritizes the selection based on the class probabilities. Each returned class accommodates a probability value between 0 and 1 within all three networks. The resultant probabilities of each returned class from all networks are summed respectively, and the highest value-based class is designated to be the final recognized class such that.

Let $P_{i}^{1}$ represent the probability of the first activity class in the ith deep learning model; then, the probabilities of the recognized activities $\left(P^{1},P^{2},....,P^{M}\right)$ in $\left(P_{1},P_{2}....,P_{n}\right)$ networks can be defined as $P_{total}^{1},P_{total}^{2},.....,P_{total}^{M}$, respectively, where *n* represents the total number of deep learning networks in the model. Then, the sum of all the probabilities against each instance can be defined as (Equation 5)
(5)$$\begin{array}{c} P_{total}^{j}=\sum_{i=1}^{n}P_{i}^{k}, \end{array}$$
where *j*=1,2,...,*M*. The cumulative probability of the same resultant classes against each instance is calculated and compared with the cumulative probability of other resultant classes. For the function f:g→j, where *g* is a subset of *j* and contains the sum of same predicted classes from each model, *j* represents the set of all generated probabilities from all networks. *k* takes the maximum argument of all the values in *g* and returns the classified activity, as shown in Equation (Equation 6).
(6)$$\begin{array}{c} g=[P_{total}^{1},P_{total}^{2},P_{total}^{3},....,P_{total}^{M}]\\ k=\underset{g}{\mathrm{argmax}}\;f\left(g\right) \end{array}$$

Finally, the class associated with the highest probability value *k* will be returned as the recognized class. The algorithm for the decision fusion module is shown in Algorithm 1.
**Algorithm 1.** Decision fusion.
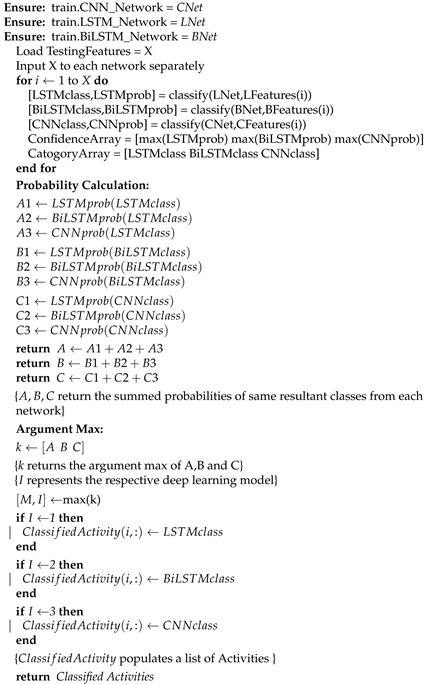


## 4. Experimental Results

### 4.1. Datasets

Two publicly available datasets were utilized for the experimental analysis of the proposed model. The following two datasets were selected based on the inclusion of transitional activities in the case of the former and to validate the performance of the proposed approach with fewer input features for the latter, respectively. Raw sensor data were transformed into a feature matrix for both datasets and then fed to the model. Next, we describe the two datasets.

#### 4.1.1. Dataset A: Human Activities and Postural Transition Dataset (HAPT)

HAPT [45] is an extended version of the UCI HAR dataset [58] and accommodates six additional postural transitions alongside six basic activities. Moreover, the dataset contains the unprocessed raw data composed of triaxial signals generated from a Samsung Galaxy II cellphone’s embedded accelerometer and gyroscope sensors. The dataset also includes the fully processed data based on a 561 feature vector, which is partitioned into two groups: one being the training input (70%) and the other being the testing input (30%).

The accumulated information is based on an experiment which involved 30 users performing 12 activities (6 basic and 6 transition). The basic activities recorded are “walking, walking upstairs, walking downstairs, sitting, standing and lying”, and the transition activities are “stand to sit, sit to stand, sit to lie, lie to sit, stand to lie and lie to stand”. An overview of the instances in dataset A is shown in Table 4.

The choice of the HAPT dataset was due to the inclusion of six postural transitions alongside basic activities. Moreover, the dataset contains fully processed data where triaxial signals have been transformed into statistical features by using a Fast Fourier Transform (FFT), including the mean, standard deviation, max, min, etc.

#### 4.1.2. Dataset B: Human Activity Dataset

This dataset [59,60] contains 24,075 observations against 5 human activities (sitting, standing, walking, running and dancing). A single observation accommodates 60 features converted from the raw triaxial data generated through a smartphone’s accelerometer and gyroscope sensor. The variables involved are as follows: *‘actid’* is a vector composed of activity IDs in the form of integers ranging from 1 to 5, *‘actnames’* is a vector composed of the activity names corresponding to their respective activity IDs, *‘feat’* is a feature vector composed of 60 features against every observation, and *‘featlabels’* is a list of names corresponding to every feature.

The decision of the selection of dataset B was based on the availability of a vast pool of observations against five basic activities. This lead to a better validation of the proposed approach on basic activities. The dataset was divided into a 90–10 proportion for training and testing, respectively. An overview of the the instances in dataset B is shown in Table 5.

### 4.2. State-of-the-Art Approaches

Table 6 shows the average recognition rate (accuracy) of various transition-aware approaches on dataset A. It very well may be noticed that the “standard deviation-based trend analysis” module fused with an SVM can achieve satisfactory results (80%) and achieve almost the same accuracy (81.62%) as the SVM infused with a transition event detection module. Both approaches are based on conventional machine learning models and fail to achieve increased performance on relatively large datasets. Comparatively, gradient-boosted decision trees outperformed (94.90%) both approaches by calculating the sum of weights from the dual training of correctly and incorrectly classified data. The approach is based on a fine-tuned SVM; however, the performance and the accuracy of the proposed approach degrades as the data are increased. This is called the curse of dimensionality, when too much data tunes the model to memorize data and cause over-fitting. Consequently, all three SVMs exhibit a common limitation corresponding to decreased performance with this increase in data. To fill the gaps, the deep learning approach utilizing multiple LSTM units, to classify transitional activities, achieved an accuracy of 89% as compared to conventional SVM approaches. However, the deep structure of LSTM with multiple networked cells slowed down the overall model, causing a vanishing gradient and increasing the computational cost. To this end, another approach utilized DBN to handle the vanishing gradient problem and extracted features from raw sensor data. The extracted features were refined by a component analysis kernel and analyzed by the LDA unit. Experimental evaluations have shown the DBNs to be superior, with 95.80% accuracy. However, DBNs have become obsolete due to their complex structure and have been replaced by a much simplified ReLu unit, which has been introduced to handle the vanishing gradient problems in neural networks.

To overcome the flaws in the afore-mentioned approaches, the CNN-LSTM approach demonstrated a pipelined model by feeding the CNN-extracted features to LSTM for refinement and feature fusion. The predicted results showed superior and equal results (95.80%) compared to the afore-mentioned state-of-the-art research works. Every approach has a significance of its own; however, the approaches lack scalability—i.e., having a complex structure—or any irregularity in the pipeline architecture can halt or slow down the system.

### 4.3. Quantitative Analysis

The quantitative analysis of the proposed framework was carried out against state-of-the-art approaches using the metrics accuracy, precision, recall and F-measure. The resultant metrics were generated based on the test data partitioned from Datasets A and B, respectively. The resultant metrics of the recognized activities from dataset A are shown in Table 7. The activity ID labels ‘‘A1,A2,A3,A4,A5,A6" represent the basic activities “Walking”, “Walking Upstairs”, “Walking Downstairs”, “Sitting”, “Standing” and “Lying”, respectively, and the labels $‘‘A7,A8,A9,A10,A11,A12^{\prime \prime }$ represent the postural transitions “Stand to Sit”, “Sit to Stand”, “Sit to Lying”, “Lying to Sit”, “Stand to Lying” and “Lying to Stand”, respectively. It can be observed that the basic activities (A1,...,A6) achieve an average precision of 97.33%, average recall of 97% and an average F1 score of 97%. However, transitional activities (A7,...,A12) showed a reduced average precision of 76.66%, average recall of 80.33% and an average F1 score of 78.16%. These results are not consistent with the results obtained for the basic activities.

The reason for this is the unavailability of abundant observations in the dataset for transition activities, causing over-fitting. Overfitting is the phenomenon of memorizing the seen data (in the case of a small dataset), meaning that the model would be unable to generalize on unseen data. Compared to basic activities, the total number of observations recorded for transition activities is significantly reduced, which caused our proposed model to over-fit. Deep neural networks perform better when the volume of data is larger; however, in this case, the volume of transitional activities varied greatly compared to basic activities, leading to less than satisfactory results. However, the final classification can be observed to have achieved the average accuracy of 96.11%, outperforming all the referenced state-of-the-art methods and portraying the overall performance of the proposed approach.

The resultant metrics of the recognized activities from dataset B are shown in Table 8. The activity ID labels ‘‘B1,B2,B3,B4,B5,B6" represent the basic activities “Sitting”, “Standing”, “Walking” “Running” and “Dancing”, respectively. It can be observed that the classified activities show an average precision of 97%, average recall of 98% and an average F1 score of 97.80%. The proposed approach showed an average accuracy of 98.38% with a higher recall, precision and F1 score compared to dataset A. The comparative analysis of basic activities in both datasets shows superior classification results for dataset B. The reason for this is the higher number of observations for individual activities in dataset B compared to dataset A. This leads to the robust training of networks in the latter case. Therefore, even though the number of features per observation was significantly greater in dataset A, results based on dataset B were superior. Table 9 shows the average accuracy of the proposed approach evaluated on the two publicly available datasets.

Table 10 shows the confusion matrix corresponding to the final classification of activities from dataset A. The diagonal bold entries represent the correctly identified instances of the activities A1,A2,...,A12, respectively. It can be observed that 490 out of 496 instances of A1 were correctly identified in the final classification; meanwhile, three instances were predicted to belong to class A2 and three from class A3. Similarly, 12 instances from A2 and 3 instances from A3 were incorrectly predicted to belong to A1. To this end, the column entries (excluding the bold entries) represent the incorrectly classified instances of those particular activities, and the row entries represent the incorrectly classified instances of the bold entries. Moreover, it can also be observed that the number of observations for transition activities was considerably smaller compared to basic activities. In a similar manner, Table 11 shows the confusion matrix relating to the final classification of activities from dataset B, where the diagonal entries represent the correctly classified instances of the activities B1,B2,...,B5.

Furthermore, a comparison of the average execution time of the proposed model was carried out with the CNN-LSTM approach on dataset A. Continuing with our explanation, we have represented 5 min (300 s) as 5 units of time. Figure 5 exhibits the difference in execution time of both approaches on 10 iterations (X-axis) where a single iteration refers to one complete execution (predictions) of each approach. Moreover, each label (0, 5, 10, ...., 60) on the Y-axis represents a difference of five units. It can be observed for the first iteration that the CNN-LSTM approach took 52 units of time for one complete execution, whereas the proposed approach took only 18 units of time while employing three deep learning models. Similarly, after 10 iterations, the average execution time for the CNN-LSTM approach is calculated to be 51.50, whereas the proposed approach demonstrates an average execution time of 17.20 units.

## 5. Conclusions

This paper proposed a hybrid multi-model framework for the efficient recognition of basic as well as transitional activities. The proposed framework utilized multiple deep learning models—i.e., LSTM, BiLSTM and CNN—followed by a decision fusion module for the final classification of activities. The proposed approach has been tested on the publicly available datasets for both basic and transition activities and compared with other state-of-the art approaches employing the same datasets. The results exhibited that the proposed approach outperformed the referenced approaches by achieving classification accuracies of 96.11% on the HAPT dataset and 98.38% on the HumanActivity dataset with transition and basic activities, respectively. For forthcoming research, the proposed approach can be transformed into a parallel architecture to further improve the processing speed for real-time implementation while putting some effort into compiling a dataset consisting of complex transition activities.

## Figures and Tables

**Figure 1 sensors-21-08227-f001:**
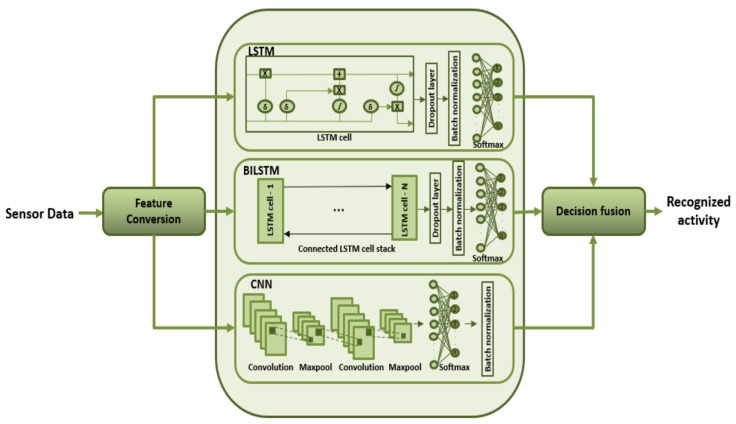
Architecture of the proposed system.

**Figure 2 sensors-21-08227-f002:**
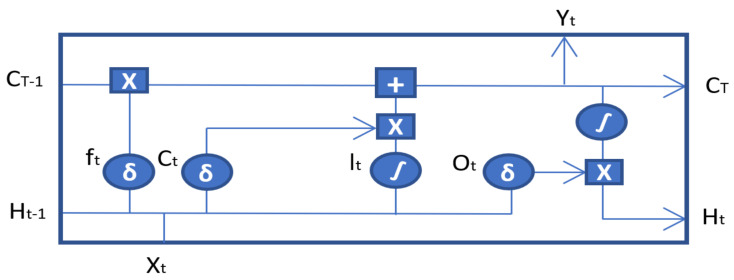
LSTM unit.

**Figure 3 sensors-21-08227-f003:**
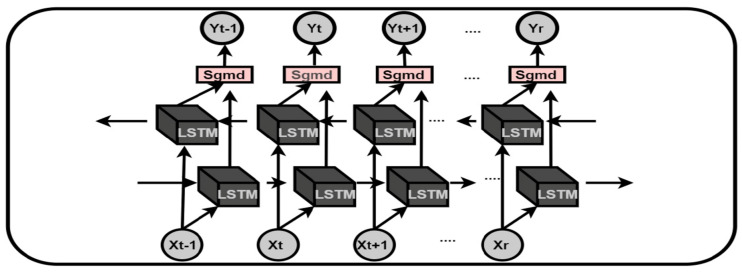
BiLSTM unit.

**Figure 4 sensors-21-08227-f004:**
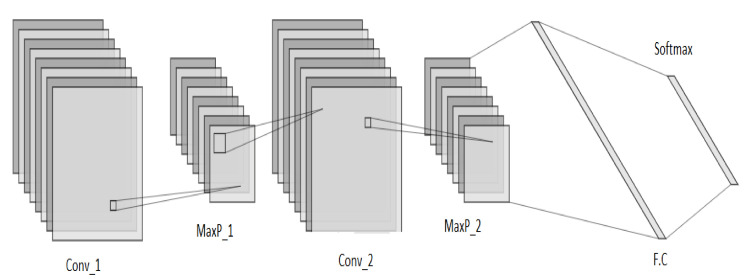
CNN unit.

**Figure 5 sensors-21-08227-f005:**
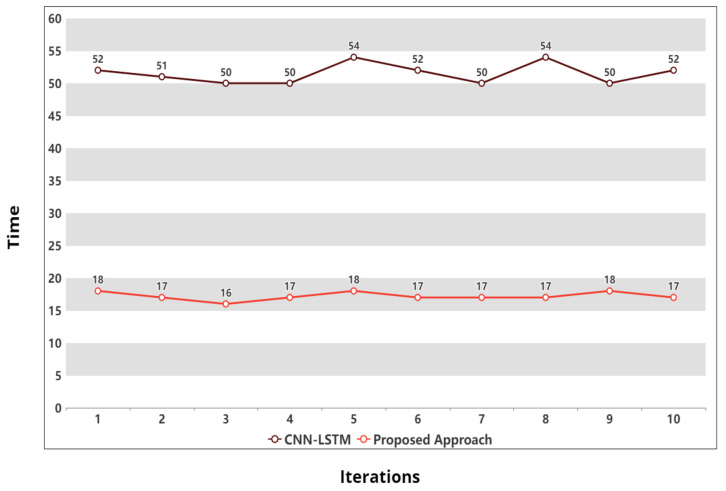
Execution time of the CNN-LSTM and proposed approach on dataset A.

**Table 1 sensors-21-08227-t001:** Literature Summary.

Ref.	Model Type	Network	Accuracy (%)	Transition Activities	Weaknesses
[32]	Machine Learning	SVM + SFFS	96.80	No	Higher accuracy on smaller datasets—increase in data causes decrease in accuracy.
[46]	Machine Learning	STD-TA	80.00	Yes	A conventional SVM with an average accuracy that extracts statistical features to differentiate between transitional and basic activities.
[47]	Machine Learning	SVM-TED	81.62	Yes	A traditional SVM with a transition event detection module to detect postural transitions but lacks accuracy for efficient identification of an action.
[40]	Deep Learning	CNN	91.00	No	Requires strongly labeled data as well as increased features in data.
[42]	Deep Learning	BiLSTM	87.50	Yes	Single BiLSTM unit cannot extract quality features from the input, no past information to correlate the data with. Works better on time series data.
[52]	Deep Learning	Multi-LSTM	89.00	Yes	Multiple pipelined LSTM units used in this approach, causing the network to train slowly and increasing the complexity of the whole model. Any fault or irregularity in a single LSTM unit affects the overall pipeline of LSTM units.
[48]	Deep Learning	DBN	95.80	Yes	DBN makes the network architecture more complex to train, and it has been replaced with ReLu, which better handles the vanishing gradient problem.
[44]	Hybrid	INN + RNN	94.00	No	INN has poor initialization, which makes it hard to debug, thus increasing the cost of the system. Moreover, a fine-tuned CNN can achieve the same or better performance than INN, which is no longer used in state-of-the-art systems.
[49]	Hybrid	GBDT	94.90	Yes	Gives best results on smaller datasets whereas accuracy decreases as the data increase.
[53]	Hybrid	CNN + LSTM	95.80	Yes	The model itself is complex and the CNN used is a conventional CNN with a basic three-layered structure that is not optimized at all. Complex activities and their transitions were not considered.

**Table 2 sensors-21-08227-t002:** LSTM & BiLSTM Parameters.

Parameter	Value-Dataset A	Value-Dataset B
*y*	561	60
ζ	0.002	0.002
η	100	50
Optimizer	Adam	Adam
κ	0.5	0.5
Epochs	400	100

**Table 3 sensors-21-08227-t003:** CNN parameters.

Parameter	Value-Dataset A	Value-Dataset B
*I*	24 × 24	8 × 8
*s*	1	1
α	8	8
β	18	18
τ	2 × 4	2 × 4
ν	2 × 8	2 × 8
γ	2	2
ϱ	ReLu	ReLu
ζ	0.002	0.002
Epochs	50	50

**Table 4 sensors-21-08227-t004:** Human Activities and Postural Transitions dataset (Dataset A) overview.

Activity	Training Instances	Test Instances
Walking	1226	496
Walking Upstairs	1073	471
Walking Downstairs	987	420
Sitting	1293	508
Standing	1423	556
Laying	1413	545
Stand to Sit	47	23
Sit to Stand	23	10
Sit to Lie	75	32
Lie to Sit	60	25
Stand to Lie	90	49
Lie to Stand	57	27

**Table 5 sensors-21-08227-t005:** Human Activity dataset (Dataset B) overview.

Activity	Training Instances	Test Instances
Sitting	5265	585
Standing	5598	622
Walking	4856	540
Running	3561	395
Dancing	2388	267

**Table 6 sensors-21-08227-t006:** Comparison with state-of-the-art approaches in terms of average accuracy (Dataset A).

Approach	Average Accuracy (%)
STD-TA [46]	80.00
SVM-TED [47]	81.62
LSTM [52]	89.00
GBDT [49]	94.90
DBN [48]	95.80
CNN-LSTM [53]	95.80
Proposed	96.11

**Table 7 sensors-21-08227-t007:** Accuracy, precision, recall and F-measure of various activities—dataset A.

Activity ID	Accuracy (%)	Precision (%)	Recall (%)	F-Measure (%)
A1	99.34	97.00	99.00	98.00
A2	99.11	98.00	96.00	97.00
A3	99.56	99.00	98.00	98.00
A4	98.13	96.00	92.00	94.00
A5	98.36	94.00	97.00	95.00
A6	100.00	100.00	100.00	100.00
A7	99.49	62.00	78.00	69.00
A8	99.97	91.00	100.00	95.00
A9	99.75	84.00	90.00	87.00
A10	99.59	73.00	76.00	75.00
A11	99.46	80.00	82.00	81.00
A12	99.46	70.00	56.00	62.00

**Table 8 sensors-21-08227-t008:** Accuracy, precision, recall and F-measure of various activities—dataset B.

Activity ID	Accuracy (%)	Precision (%)	Recall (%)	F-Measure (%)
B1	99.92	100.00	100.00	100.00
B2	99.88	100.00	100.00	100.00
B3	99.58	99.00	99.00	99.00
B4	98.71	96.00	96.00	96.00
B5	98.67	93.00	95.00	94.00

**Table 9 sensors-21-08227-t009:** Average accuracy of the proposed approach on two datasets.

	Average Accuracy (%)	
Proposed Approach	dataset A	dataset B
	96.11%	98.38%

**Table 10 sensors-21-08227-t010:** Confusion matrix of activities—dataset A.

		Predicted											
		**A1**	**A2**	**A3**	**A4**	**A5**	**A6**	**A7**	**A8**	**A9**	**A10**	**A11**	**A12**
Actual	A1	490	12	3	0	0	0	0	0	0	0	0	0
	A2	3	454	7	0	0	0	1	0	0	0	0	0
	A3	3	1	410	0	0	0	0	0	0	0	0	0
	A4	0	0	0	467	15	0	2	0	0	0	1	0
	A5	0	0	0	35	540	0	1	0	0	0	0	0
	A6	0	0	0	0	0	545	0	0	0	0	0	0
	A7	0	4	0	6	1	0	18	0	0	0	0	0
	A8	0	0	0	0	0	0	1	10	0	0	0	0
	A9	0	0	0	0	0	0	0	0	27	0	5	0
	A10	0	0	0	0	0	0	0	0	0	19	2	5
	A11	0	0	0	0	0	0	0	0	3	0	36	6
	A12	0	0	0	0	0	0	0	0	0	6	0	14

**Table 11 sensors-21-08227-t011:** Confusion matrix of activities—dataset B.

		Predicted				
		**A1**	**A2**	**A3**	**A4**	**A5**
Actual	A1	584	1	0	0	0
	A2	0	619	7	0	0
	A3	1	2	536	3	0
	A4	0	0	0	378	14
	A5	0	0	4	14	251

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
