# Peer review of "A Novel Hybrid Deep Learning Model for Human Activity Recognition Based on Transitional Activities"

_sensors, 2021, doi:10.3390/s21248227_

Round 1
Reviewer 1 Report
The authors present a hybrid deep learning model for human activity recognition (HAR) that is designed to cope with both basic and transitional activities. They evaluate their method empirically on two publicly available HAR data-sets. Of these, one consists of both transitional and basic activities, and the other only of basic activities. While the authors present results for the proposed method and the state-of-the-art on one data-set, it is unclear which of the two data-sets they correspond to. If they correspond to the data-set that is limited to basic activities, then these results say nothing about whether or not, and to what degree, the proposed method succeeds in achieving the goal of improving performance for transitional activities.
Either way, the performance improvements on the state-of-the-art are not that big (0.31 percentage points better than other DL methods, and 1.21 percentage points better than non-DL ML methods). To be convinced of the proposed approach’s superiority over these other approaches, I would need to see comparable results for multiple independent data-sets. Ideally, this would be the accuracy when each method is applied to each of multiple (≥ 5) data-sets. At a minimum, it would be the accuracy when the proposed method is applied to multiple data-sets. The two data-points the paper offers is nowhere near enough.
All that said, the paper is enough to show that your method performs better than the state-of-the-art on at least one dataset, and the results for both types (transitional and basic) activities are useful data-points for researchers in the area of transitional HAR.
Below follow a list of items I want to see addressed before I recommend the paper for publication.
- I am assuming that all the results (Average Accuracies, Confusion matrices, etc.) are for the testing data? Please clarify.
- Section 4.2: “Table 4 shows the average recognition…” Please clarify what data-set(s) these results refer to. The accuracy value for the “Proposed” method (96.11) suggests it is data-set A, but on closer inspection of Figure 3 (which claims to be for data-set B)
- Also, please clarify how you obtained the results for the state-of-the-art approaches in Table 4. Are they taken from the literature, or did you implement some (which ones?) or all of them yourself?
- Subsections 4.1.1 and 4.1.2: Please summarize (in a table or graphically) the number (and, if possible, percentage) of samples per activity for each of the two data-sets. This will help readers better put the results in context (mentally account for class imbalances). For Dataset A (4.1.1), please also state the size/ percentage of both training and testing data.
- Tables 8 and 9: The column labels (A1, A2, …, and B1, B2, …) are not aligned properly. Please correct.
- I am a bit confused by your presentation of results. Table 7 lists the average accuracies of the proposed approach for data-set A and B as 96.11% and 98.38%, respectively. Then how is it that Figure 3, whose caption clearly associates it with dataset B, shows the proposed approach’s accuracy as 96.11% and not 98.38%? Moreover, the numbers from this figure are clearly repeated from Table 4. If you really need both representations (graphical and tabular), then it would be better to present them together—and you have to clarify, in a consistent way, which data-sets all of these numbers refer to.
- Figure 2, and associated text. I presume “execution time” refers to testing (i.e., prediction), not training, but you mentioning “iterations” has me a unsure whether this is what you meant? Please clarify what you mean by “execution time.”
Author Response
A point-by-point response to the comments is attached.

Reviewer 2 Report
The authors propose a hybrid multi-model activity recognition approach using deep learning models. The idea sounds interesting, the main highlighted points follow such as:
1) The Abstract Section needs to describe a brief summarization of quantitative results.
2) Section 1 presents the motivation and the problem under study. This section is very well written and organized. Nowadays, there is an interesting contribution of the literature using Deep Learning when deployed into embedded devices. This new being is called TinyML. The authors are invited to add some discussion regarding TinyML when applied to the envisaged problem.
3) Section 2 is a bit interesting; all papers are compared using a very didact Table. However, there is a limitation in the Table because it is not mentioned the computational demand of all solutions. Indeed, solutions should go beyond a prototype, and deployment must be a choice by design.
4) After reading Section 3, the most impressive of that section is a revision of the literature. Contributions are limited only to section 3.5. The authors need to clarifty this point.
5) Resolution of Figure 2 need to be improved. Unit of axis Y is missing.
6) Authors need to share a public repository with all source code and datasets used to make the results reproducible. The pipeline used by the authors is also a contribution, and it should be shared.
7) Conclusion is timid given the enormous amount of results presented by the authors should expand this section.
Author Response

(The authors gave the same response as above.)

Round 2
Reviewer 2 Report
The authors implemented all suggestions previously mentioned. Open issues were solved, thus the reviewer suggests accept the paper in the present form.